# The Distribution of the Genotypes of *ABCB1* and *CES1* Polymorphisms in Kazakhstani Patients with Atrial Fibrillation Treated with DOAC

**DOI:** 10.3390/genes14061192

**Published:** 2023-05-29

**Authors:** Ayan Abdrakhmanov, Ainur Akilzhanova, Aizhan Shaimerdinova, Madina Zhalbinova, Gulnara Tuyakova, Svetlana Abildinova, Rustam Albayev, Bayan Ainabekova, Assel Chinybayeva, Zhanasyl Suleimen, Makhabbat Bekbossynova

**Affiliations:** 1National Research Cardiac Surgery Center, Astana 010000, Kazakhstan; ayan-3@mail.ru (A.A.);; 2Medical Centre Hospital of President’s Affairs Administration of the Republic of Kazakhstan, Astana 010000, Kazakhstan; albaev78@bk.ru; 3Center for Life Sciences, National Laboratory Astana, Nazarbayev University, Astana 010000, Kazakhstan; madina.zhalbinova@nu.edu.kz; 4Department of Internal Medicine, Medical University Astana, Astana 010000, Kazakhstan; ai12-10@mail.ru (A.S.);; 5Corporate Fund “University Medical Center”, Nazarbayev University, Astana 010000, Kazakhstan; chena@k.ru

**Keywords:** dabigatran, apixaban, *ABCB1*, *CES1*, pharmacogenetics, NVAF

## Abstract

Nowadays, direct oral anticoagulants (DOACs) are the first-line anticoagulant strategy in patients with non-valvular atrial fibrillation (NVAF). We aimed to identify the influence of polymorphisms of the genes encoding P-glycoprotein (*ABCB1*) and carboxylesterase 1 (*CES1*) on the variability of plasma concentrations of DOACs in Kazakhstani patients with NVAF. We analyzed polymorphisms rs4148738, rs1045642, rs2032582 and rs1128503 in *ABCB1* and rs8192935, rs2244613 and rs71647871 *CES1* genes and measured the plasma concentrations of dabigatran/apixaban and biochemical parameters in 150 Kazakhstani NVAF patients. Polymorphism rs8192935 in the *CES1* gene (*p* = 0.04), BMI (*p* = 0.01) and APTT level (*p* = 0.01) were statistically significant independent factors of trough plasma concentration of dabigatran. In contrast, polymorphisms rs4148738, rs1045642, rs2032582 and rs1128503 in *ABCB1* and rs8192935, rs2244613 and rs71647871 *CES1* genes did not show significant influence on plasma concentrations of dabigatran/apixaban drugs (*p* > 0.05). Patients with GG genotype (138.8 ± 100.1 ng/mL) had higher peak plasma concentration of dabigatran than with AA genotype (100.9 ± 59.6 ng/mL) and AG genotype (98.7 ± 72.3 ng/mL) (Kruskal–Wallis test, *p* = 0.25). Thus, *CES1* rs8192935 is significantly associated with plasma concentrations of dabigatran in Kazakhstani NVAF patients (*p* < 0.05). The level of the plasma concentration shows that biotransformation of the dabigatran processed faster in individual carriers of GG genotype rs8192935 in the *CES1* gene than with AA genotype.

## 1. Introduction

Atrial fibrillation (AF) is one of the important problems in public health which affects the aging population (>65) worldwide. AF affects ≥1% of the aging population in Australia, Europe and the US [1,2]. Patients with nonvalvular AF (NVAF) have lower cardiac output due to the mechanical dissociation of atrial and ventricular contraction [2]. NVAF patients are at higher risk of developing stroke, heart failure, myocardial infarction, chronic kidney diseases, dementia, high mortality rate and other cardiovascular diseases [1,2].

Patients with NVAF are usually prescribed anticoagulant or antiplatelet therapy to reduce the risk of thromboembolic stroke [3]. Warfarin is one of the basic anticoagulant therapies which indirectly inhibits activation of the coagulation factors II, VII, IX and X by inhibition of vitamin K epoxide reductase complex 1 (VKORC1) [2,4]. However, not every patient is treated with warfarin due to the frequent control of international normalized ratio (INR) range, higher/lower dosing, compliance with dose adjustments and possible risks of bleeding events [2,3].

Nowadays, direct oral anticoagulants (DOACs) are the first-line anticoagulant strategy in patients with non-valvular atrial fibrillation (NVAF). DOACs are the first alternative to vitamin K antagonists (VKAs). DOACs are targeted inhibitors which deactivate the final pathway of the coagulation cascade, whereas warfarin inhibits multiple coagulation factors [2]. Dabigatran and apixaban are direct inhibitors which deactivate thrombin and coagulation factor Xa in the coagulation cascade. These coagulation factors have important roles in the coagulation process of the hemostasis [2,5].

Dabigatran etexilate is an oral prodrug which is converted into the active form of dabigatran by the liver carboxylesterase 1 (CES1) enzyme [6]. Bioavailability of dabigatran etexilate occurs by P-glycoprotein, an adenosine triphosphate (ATP)-dependent drug efflux transporter which is encoded by the *ABCB1* gene [6,7]. On the other hand, apixaban is recommended in older patients with increased risks of bleeding, thromboembolism complications and moderate renal dysfunction [6]. It is metabolized by cytochrome P450 (CYP3A4/5) and transported by the P-glycoprotein (ABCB1) and BCRP (breast cancer resistance protein) transporters [7,8].

DOAC’s activation, transport, metabolism and concentration of their active metabolites differentiates in various individuals. However, their dosage is prescribed in a fixed amount (dabigatran 110 mg and 150 mg; apixaban 2.5 mg and 5 mg;) unlike warfarin, with different dosage. Concentration variability in patients might cause development of thromboembolic or bleeding complications during treatment [9,10].

The higher or lower concentrations of active metabolites depend on the genotype polymorphisms of *ABCB1* and *CES1* genes in different individuals [11]. Genome-wide association studies investigated influence of polymorphisms of *ABCB1* and *CES1* genes on dabigatran concentrations [11,12]. Research concludes that lower concentration is associated with polymorphism rs2244613 in the *CES1* gene whereas higher concentration is associated with polymorphisms rs4148738 in *ABCB1* and rs8192935 in *CES1* genes [12]. Concentration of active metabolites was shown to be different according to the genotype frequency of single-nucleotide polymorphisms (SNP) of *ABCB1* and *CES1* genes in different populations such as European, Chinese, Caucasian and Russian [10,11,12,13].

However, there has been no study performed on association analysis of *ABCB1* and *CES1* genes in NVAF patients with prescribed DOAC treatments in the Kazakhstani population [14,15]. Thus, the aim of the study was to identify the distribution and influence of polymorphisms of the genes encoding P-glycoprotein (*ABCB1*) and carboxylesterase 1 (*CES1*) on the variability of plasma concentrations of DOACs in Kazakhstani patients with NVAF.

## 2. Materials and Methods

### 2.1. Study Participants

The study was conducted in accordance with the Declaration of Helsinki and approved by the Ethics Committee of the National Research Cardiac Surgery Center (NRCSC), Astana (No. 01-74 from 10 June 2021) for studies involving humans. Written informed consent was obtained from all patients. To protect patients’ rights and not disclose their personal information, the database was encoded and depersonalized.

We recruited 150 Kazakh patients (90 males and 60 females, 58.9 ± 10.3 years old) with NVAF during August 2021–July 2022. Basic clinical characteristics were collected from medical records. CHA_2_DS_2_-VASc and HAS–BLED scores were calculated for every NVAF patient (Table 1). Exclusion criteria for participation in the study were: chronic renal failure, chronic liver disease, valvular atrial fibrillation, cancer, history of peptic ulcer, allergic reactions to the study drugs, age over 75 years and refusal to participate in the study.

All patients were sequentially prescribed 150 mg of dabigatran (Pradaxa, Boehringer-Ingelheim, Ingelheim am Rhein, Germany) and 5 mg of apixaban (Eliquis, Pfizer, New York, NY, USA) twice a day to determine sensitivity to two anticoagulants. To identify clear plasma concentration levels of active metabolites (dabigatran, apixaban), patients did not take interacting drugs that are inducers or inhibitors of p-gp. Venous blood samples were collected for the measurement of the plasma concentrations of dabigatran/apixaban and biochemical parameters. Furthermore, venous blood samples were obtained into sterile vacutainers with K_2_EDTA for genotyping analysis.

Patients were categorized into two groups according to their gender for comparative analysis: Male, *n* = 90 (Group 1); Female, *n* = 60 (Group 2). Basic clinical characteristics, biochemical parameters and plasma concentrations of dabigatran/apixaban drugs were compared between gender groups.

Furthermore, patients were stratified by age for comparative analysis: *n* = 100, younger age: <65 years (Group 1); *n* = 50, older age ≥ 65 years (Group 2). Basic clinical characteristics, biochemical parameters and plasma concentrations of dabigatran/apixaban drugs were compared between these two groups.

### 2.2. Determination of Plasma Concentration of DOAC

Plasma concentrations of dabigatran and apixaban were measured by “AST-TOP 500”, Instrumentation laboratory, USA. Plasma concentrations of drugs were identified in two parts. The 1st part was focused on determination of plasma concentration of apixaban whereas the 2nd part was performed on identification of plasma concentration of dabigatran. Venous blood samples were collected within 1–3 h after intake of dabigatran (150 mg) for measurement of peak concentration. Then, the trough concentration was measured after 10–11 h. Furthermore, plasma concentration of apixaban was measured after intake (after 48 h). Venous blood samples were obtained within 1–2 h after intake of apixaban (5 mg) for measurement of the peak concentration. Then, the trough concentration level was measured after 11 h.

### 2.3. DNA Isolation and SNP Genotyping

Genomic DNA was isolated from 200 µL of venous blood samples by using Illustra blood genomic Prep Mini Spin Kit (Illustra, Nottingham, UK) according to the protocol instructions. Extracted DNA’s concentration and purity were determined by NanoDrop™ Spectrophotometer (ThermoFisher Scientific, Waltham, MA, USA). Consequently, genomic DNA samples were genotyped for seven SNPs by using real-time polymerase chain reaction (qPCR) with allele discrimination using TaqMan Real Time PCR Assay on a 7900HT Fast Real-Time PCR System (Applied Biosystems, Waltham, MA, USA). We studied four SNPs in *ABCB1* (rs4148738, rs1045642, rs2032582 and rs1128503) and three in *CES1* (rs8192935, rs2244613 and rs71647871) genes encoding P-glycoprotein transporter and liver carboxylesterase 1 (CES1) enzyme.

### 2.4. Statistical Analysis

Data of continuous variables were presented as mean ± standard deviation (SD). Normality of distribution of continuous variables was identified by Kolmogorov–Smirnov test (*p* > 0.05). Comparison analysis of continuous variables between two groups was assessed by Student’s *t*-test (normally distributed variables) and by non-parametric Mann–Whitney U test (not normally distributed variables). Furthermore, continuous variables between three genotypes of SNPs were compared by non-parametric Kruskal–Wallis test (not normally distributed variables) and by one-way ANOVA (normally distributed variables). Categorical variables were presented as frequencies and percentages which were compared using chi-square test or Fisher’s exact test. Power analysis and sample size calculation were assessed on online calculator https://clincalc.com (assessed on 4 March 2023). The sample size in the gender groups 60% (n = 90, male) and 40% (n = 60, female) achieved 85% of power with an α value of 0.05. Hardy–Weinberg equilibrium (HWE) for genotype frequencies was processed using chi-square test or Fisher’s exact test. Odds ratios (OR) with 95% confidence interval (CI) and p value assessed associations between polymorphisms and gender groups. Logistic regression analysis was processed to evaluate difference in genotype distributions using web tool https://snpstats.net/ (assessed on 6 February 2023).

Furthermore, we performed multiple linear regression models to identify relationships between plasma concentration of dabigatran/apixaban and basic clinical, biochemical parameters, polymorphisms of rs4148738, rs1045642, rs2032582 and rs1128503 in the *ABCB1* gene and rs8192935, rs2244613 and rs71647871 *CES1* gene.

In addition, multinomial logistic regression analysis was performed to identify relationships between genotype polymorphism of rs8192935 in the CES1 gene and clinical, biochemical parameters, peak plasma concentrations of dabigatran and apixaban. Statistical analysis was processed in SPSS program version 23 (SPSS, Chicago, IL, USA).

## 3. Results

### 3.1. Analysis of Clinical, Biochemical and Pharmacokinetics Parameters

Basic clinical characteristics of NVAF patients and their comparative analysis between gender and age groups are summarized in Table 1. Patients were diagnosed with paroxysmal (44%) and persistent (56%) types of AF. Age, height, weight, INR level, smoking status and diabetes were significantly different between gender and age groups (*p* < 0.05). During follow up period in our study (2021–2022) no complications such as thromboembolism and bleeding were developed.

In our patient cohort, CHA_2_DS_2_-VASc scores ranged from 0 to 7. Fifty-four (60%) patients in the male group had score 1 whereas twenty-three (38.3%) female patients had the same score, which showed significantly less percentage (*p* = 0.001). On the other hand, HAS–BLED scores ranged from 0 to 3 without significance.

CHA_2_DS_2_-VASc score 1 frequency showed significantly increased numbers in the younger age group < 65 than in the group of patients older than age ≥ 65 (*p* = 0.002). Furthermore, score frequency of HAS–BLED exhibited significant differences between age groups (*p* = 0.002).

Arterial hypertension and regurgitation events were more prevalent in the group of patients of older age ≥ 65. Forty-six (92%) patients had arterial hypertension and thirty-nine (78%) had regurgitation in the group of older patients (age ≥ 65), (*p* < 0.05 for both conditions).

Biochemical parameters and plasma concentrations of dabigatran/apixaban of NVAF patients and their comparative analysis between gender and age groups are summarized in Table 2. Male patients had significantly higher level of hemoglobin and creatinine than females (*p* < 0.05). On the contrary, fibrinogen level was significantly higher in females than in males (*p* = 0.01). Other biochemical parameters did not show any statistically significant variations between gender groups (Table 2). Comparative analysis showed significant differences of plasma concentrations of dabigatran/apixaban drugs between male and female groups (*p* < 0.05). Peak dabigatran concentration was significantly higher in the female group than in the male group (122.8 ± 73.3 vs. 92.4 ± 70.3 ng/mL, *p* = 0.003). Peak apixaban concentration level was higher in females than in males with almost significant results (181.4 ± 88.2 vs. 153.9 ± 68.9, respectively, *p* = 0.06). Trough dabigatran and apixaban concentrations were significantly higher in the female group than in the male group (*p* < 0.05).

The group of younger patients (age < 65) had significantly higher levels of hemoglobin, erythrocytes and platelets than the group of patients of older age ≥ 65 (*p* < 0.05). Other biochemical parameters did not show any statistically significant variations between age groups (Table 2).

Plasma concentrations of dabigatran/apixaban showed significant differences between groups of patients with age (*p* < 0.05). Trough dabigatran concentration was significantly higher in the group of patients of age ≥ 65 than in the group of patients younger than 65 (59.6 ± 32.3 vs. 53.6 ± 40.5, *p* = 0.05). However, peak concentration of dabigatran was not significantly different between these groups (*p* = 0.60). Peak apixaban concentration was significantly higher in the group of patients of older age ≥ 65, than in the group of patients of younger age < 65 (181.3 ± 85.3 vs. 156.8 ± 73.4, *p* = 0.05). Trough apixaban concentration was significantly higher in the group of older patients, age ≥ 65, than in the group of younger patients, age < 65 (115.7 ± 50.1 vs. 93.8 ± 42.1, respectively, *p* = 0.002).

### 3.2. Analysis of Genotyping

Seven polymorphisms of rs4148738, rs1045642, rs2032582 and rs1128503 in *ABCB1* and rs8192935, rs2244613 and rs71647871 in *CES1* genes were genotyped in NVAF patients (*n* = 150). The distributions of genotype and allelic frequencies in NVAF patients and in other populations such as European, African, African-American, Asian and East Asian were summarized in Table 3. The distributions of allelic frequencies in other populations were taken from the database of https://www.ncbi.nlm.nih.gov/ (assessed on 5 March 2023).

Our investigation found that the distributions of allelic frequencies of SNPs in the *ABCB1* and *CES1* genes of Kazakhstani population is slightly similar to Asian and East Asian populations (Table 3). However, some of the polymorphisms showed difference in allele frequency in Asian population. For instance, the frequency of G allele polymorphism rs2244613 in the *CES1* gene in Kazakhstani population (0.48) was slightly lower than in Asian (0.57) and East Asian populations (0.61). Moreover, frequency of alleles in polymorphisms of rs4148738, rs2032582 and rs1128503 in the *ABCB1* gene in the Kazakhstani population was identified to be similar with European population (Table 3). The allele frequency of African and African-American populations is totally different from the Kazakhstani population. For instance, frequency of C allele in polymorphism rs4148738 in the *ABCB1* gene was higher (0.43) compared to the African-American (0.24) population. Interestingly frequency of A allele polymorphism rs71647871 in the *CES1* gene was almost absent in previously listed populations (Table 3).

Furthermore, we analyzed distributions of allelic and genotype frequencies among gender group due to the identification significant difference in plasma concentrations of dabigatran and apixaban drugs (Table 4).

The distributions of allelic frequencies were tested for Hardy–Weinberg equilibrium (HWE) between gender groups (Table 4). The distributions of allelic and genotype frequencies of polymorphisms rs4148738, rs1045642, rs2032582 and rs1128503 in *ABCB1* and rs8192935, rs2244613 and rs71647871 in *CES1* genes were not significantly different between male and female groups (Table 4) (*p* > 0.05). Furthermore, logistic regression analysis (adjusted age and BMI) performed that polymorphisms rs4148738, rs1045642, rs2032582 and rs1128503 in *ABCB1* and rs8192935, rs2244613, rs71647871 in *CES1* genes are not significantly associated in NVAF patients’ gender groups (*p* > 0.05). Results of logistic regression analysis are summarized in Appendix A.

Additionally, we performed comparative analysis of plasma concentrations of dabigatran and apixaban between patients carrying different genotypes of polymorphisms rs4148738, rs1045642, rs2032582 and rs1128503 in *ABCB1* and rs8192935, rs2244613 and rs71647871 in *CES1* genes to identify influence of inherited information in NVAF patients (Table 5). Comparative analysis of plasma concentrations did not show significant difference between genotype polymorphisms among gender groups (*p* > 0.05).

### 3.3. Multiple Linear Regression Analysis

Multiple linear regression analysis was aimed to analyze influence of the basic clinical, biochemical parameters, polymorphisms of rs4148738, rs1045642, rs2032582 and rs1128503 in the *ABCB1* gene and rs8192935, rs2244613, rs71647871 in the *CES1* gene to the peak and trough plasma concentration level of dabigatran/apixaban in AF patients.

Linear regression analysis identified polymorphism rs8192935 in the *CES1* gene (95% CI: 0.26–47.7, *p* = 0.05), gender (95% CI: 1.52–56.1, *p* = 0.04) and APTT level (95% CI: 0.64–4.47, *p* = 0.009) are statistically significant independent factors of peak plasma concentration of dabigatran (Table 6).

On the other hand, polymorphism rs8192935 in the *CES1* gene (95% CI: 0.91–24.7, *p* = 0.04), BMI (95% CI: −4.05–0.67, *p* = 0.01) and APTT level (95% CI: 0.84–2.76, *p* = 0.01) are statistically significant independent factors of trough plasma concentration of dabigatran (Appendix A). Our investigation found that these listed independent factors are predictors of the peak and trough plasma concentration of the dabigatran drug.

On the contrary, gene polymorphisms were not identified to be significant predictors of the plasma concentration of apixaban (*p* > 0.05). However, age (95% CI: 0.14–1.69, *p* = 0.02), BMI (95% CI:–5.11–0.95, *p* = 0.01) and fibrinogen (95% CI: 0.74–16.7, *p* = 0.03) were statistically significant independent factors of trough plasma concentration of apixaban (Appendix A).

### 3.4. Multinomial Logistic Regression Analysis

Multinomial logistic regression analysis aimed to identify relationships between genotypes of polymorphism rs8192935 in the *CES1* gene and age, BMI, APTT level, fibrinogen, peak plasma concentrations of dabigatran and apixaban.

Regression analysis showed that peak plasma concentration of dabigatran was higher in the presence of the reference genotype (GG) polymorphism of rs8192935 in the *CES1* gene than with AA (OR 0.99, 95% CI: 0.98–1.00, *p* = 0.04) and AG (OR 0.99, 95% CI: 0.99–1.00, *p* = 0.03) genotypes (Appendix A).

## 4. Discussion

Our study aimed to identify genotype distributions of polymorphisms which influence the transport of the dabigatran/apixaban drugs and convert them into active form in AF patients with prescribed anticoagulant treatment. We analyzed polymorphisms of rs4148738, rs1045642, rs2032582 and rs1128503 in *ABCB1* and rs8192935, rs2244613 and rs71647871 in *CES1*, genes which were investigated previously in different populations in relation to dabigatran/apixaban treatments [11,13,16]. These polymorphisms were not investigated in the Kazakhstani population previously [14,15]. In our study, we found that polymorphism rs8192935 in the *CES1* gene is significantly associated with plasma concentrations of dabigatran in Kazakhstani population (*p* < 0.05).

We could not estimate changes in clinical outcomes associated with pharmacogenetic testing in our study because no serious complications during dabigatran/apixaban treatments intake were observed as we only had a one-year follow-up period (2021–2022). We compared plasma concentrations of dabigatran/apixaban drugs between groups divided by gender and age (Table 2). We found that females had significantly higher level of peak and trough plasma concentrations of dabigatran/apixaban than males (*p* < 0.05). On the other hand, higher levels of peak and trough plasma concentrations of dabigatran/apixaban were significantly higher in the older (age ≥ 65) group than in the younger (age < 65) group (*p* < 0.05). Investigation of the RE-LY trial revealed that patient’s age, weight, creatinine clearance and gender are significant factors which influence the level of dabigatran plasma concentrations [10,17]. Our study identified these factors to be significant too (*p* < 0.05). Moreover, biochemical parameters were also compared and showed significant difference between gender and age groups (Table 2). Biochemical parameters such as hemoglobin, erythrocytes, platelets, creatinine and fibrinogen were significantly different between gender and age groups (*p* < 0.05).

Our study was the first study in Kazakhstani patients with AF to identify genotype distribution of the polymorphisms in *ABCB1* and *CES1* genes and included case series only. We compared the distributions of allelic frequencies of seven SNPs in the *ABCB1* and *CES1* genes with European, African, African-American, Asian and East Asian populations (Table 3). We observed that the distributions of allelic frequencies of SNPs in the *ABCB1* and *CES1* genes of Kazakhstani population was similar to Asian and East Asian populations. We did not identify significant differences in the distribution of allelic and genotype frequencies of polymorphisms between gender groups (Table 4).

Furthermore, peak and trough plasma concentrations of dabigatran/apixaban were compared between genotype polymorphisms (Table 5). In the comparative analysis, polymorphisms of rs4148738, rs1045642, rs2032582 and rs1128503 in *ABCB1* and rs8192935, rs2244613, rs71647871 in *CES1* did not show significant influence on plasma concentrations of dabigatran/apixaban drugs (*p* > 0.05). However, polymorphism rs2032582 in the *ABCB1* gene showed slightly significant results between genotypes (*p* = 0.09). Ueshima et al. (2017) in their investigation identified that Japanese patients with non-valvular AF had slightly higher concentration of apixaban with presence of the CC genotype in polymorphism rs2032582 in the *ABCB1* gene than with CA genotype [8]. On the contrary, our study showed that AF patients with the presence of the AA genotype for polymorphism rs2032582 in the *ABCB1* gene had higher level of plasma concentration of apixaban than with heterozygote CA genotype (111.9 ± 48.2 ng/mL vs. 91.5 ± 41.7 ng/mL, *p* = 0.09).

Even though our study did not show any significant results in distribution of allelic and genotype frequencies, we identified that polymorphism rs8192935 in the *CES1* gene could be a predictor of the peak (95%CI: 0.26–47.7, *p* = 0.05) and trough (95% CI: 0.91–24.7, *p =* 0.04) plasma concentrations of dabigatran (Table 6 and Appendix A). Dimatteo et al. (2016) [10] showed previously that the polymorphism rs8192935 in the *CES1* gene significantly influenced trough plasma concentration of dabigatran. However, their research did not find polymorphism rs8192935 in the *CES1* gene to be significantly associated with dabigatran’s peak plasma concentrations. So, patients with GG genotype had significantly higher trough concentration (85.4 ng/dL) than with AA genotype (53.5 ng/dL) (*p* < 0.05) [10]. On the contrary, in our study polymorphism rs8192935 in the *CES1* gene was significantly associated with peak and trough plasma concentrations of dabigatran according to the linear regression analysis (Table 6 and Appendix A). Our comparative analysis of peak and trough plasma concentrations between genotype polymorphisms were not proved to be significantly different (Table 5) (Kruskal–Wallis test, *p* > 0.05). However, additional analysis of multinomial logistic regression confirmed that genotype polymorphism rs8192935 in the *CES1* gene might have significant influence on the peak plasma concentration of dabigatran and difference between genotypes (Appendix A). As *CES1* enzyme causes biotransformation of the dabigatran into its active form, its genetic variation might cause variability in the plasma concentration of dabigatran [10]. Our study showed that patients with GG genotype (138.8 ± 100.1 ng/mL) had higher peak plasma concentration of dabigatran than with AA genotype (100.9 ± 59.6 ng/mL), and AG genotype (98.7 ± 72.3 ng/mL) (Kruskal–Wallis test, *p* = 0.25). According to the results, the level of the plasma concentration shows that biotransformation of dabigatran processed faster in individuals with presence of the GG genotype of polymorphism rs8192935 in the *CES1* gene than with AA genotype. Polymorphism rs8192935 in the *CES1* gene was found to be associated with peak plasma concentration and without associations with bleeding, thromboembolic and ischemic events [9,12].

There are several limitations in the present study. First, this study was a single-center, prospective study with a small subject cohort, case series only study. We therefore could not perform multivariable analyses adjusting for a sufficient number of confounding factors. Second, we could follow up our cases after a one-year period. We could not estimate changes in clinical outcomes associated with pharmacogenetic testing in our study because no serious complications during dabigatran/apixaban intake were observed during the follow up period. Third, the lack of a population control group limited us to identify significant differences in the distributions of the allelic and genotype frequencies of genetic polymorphisms. Finally, our results have yet to be verified in an external validation cohort, which limits the generalizability of our results.

## 5. Conclusions

Our study showed that polymorphism rs8192935 in the *CES1* gene was significantly associated with plasma concentrations of dabigatran in Kazakhstani NVAF patients. According to the results, the level of the plasma concentration shows that biotransformation of the dabigatran processed faster in individuals with presence of the GG genotype of polymorphism rs8192935 in the *CES1* gene than with AA genotype. Larger studies are needed to confirm our observations. Anticoagulant treatment of dabigatran with apixaban did not show serious complications in Kazakhstani NVAF patients during a one-year follow up in our study. Pharmacogenetic testing could help to prescribe appropriate dosages of the DOAC treatment for NVAF patients, which will increase efficacy of the therapy. However, further evaluation is needed to clarify the relationship between genotypes and DOAC treatment with long-term observation of clinical outcomes in larger cohort of AF patients. Large population-based screening studies are also needed to establish the frequency and significance of the variations in the sequence of genes in the Kazakhstani population.

## Figures and Tables

**Table 1 genes-14-01192-t001:** Basic clinical characteristics of NVAF patients and comparison between gender and age groups.

Characteristics	NVAF Patients, *n* = 150	Comparison between Gender			Comparison between Age Groups		
		Male, *n* = 90	Female, *n* = 60	*p* Value	Age < 65, *n* = 100	Age ≥ 65, *n* = 50	*p* Value
Age (years)	58.9 ± 10.3	56.7 ± 11.3	62.1 ± 7.52	0.002 *	54.2 ± 9.41	68.2 ± 2.67	0.001 *
Height (cm)	166.1 ± 9.90	171.7 ± 7.11	157.7 ± 7.18	0.001 **	168.8 ± 9.43	160.6 ± 8.52	0.001 *
Weight (kg)	78.1 ± 13.8	83.6 ± 12.4	69.8 ± 11.5	0.001 **	80.2 ± 14.2	73.7 ± 11.7	0.005 **
BMI (kg/m)	28.2 ± 3.65	28.3 ± 3.24	28.1 ± 4.20	0.73	28.1 ± 3.87	28.5 ± 3.17	0.53
Type of atrial fibrillation							
paroxysmal	66 (44.0)	34 (37.8)	32 (53.3)	0.07	44 (44.0)	22 (44.0)	1
persistent	84 (56.0)	56 (62.2)	28 (46.7)		56 (56.0)	28 (56.0)	
INR	1.04 ± 0.21	1.03 ± 0.19	1.07 ± 0.25	0.5	1.02 ± 0.17	1.10 ± 0.27	0.03*
CHA2DS2-VASc, score							
0	11 (7.3)	10 (11.1)	1 (1.7)	0.001	11 (11.0)	0 (0.0)	0.002
1	77 (51.3)	54 (60.0)	23 (38.3)		60 (60.0)	17 (34.0)	
2	37 (24.7)	18 (20.0)	19 (31.7)		21 (21.0)	16 (32.0)	
3	22 (14.7)	8 (8.9)	14 (23.3)		7 (7.0)	15 (30.0)	
4	2 (1.3)	0 (0)	2 (3.3)		1 (1.0)	1 (2.0)	
7	1 (0.7)	0 (0)	1 (1.7)		0 (0.0)	1 (2.0)	
HAS–BLED, score							
0	24 (16.0)	15 (16.7)	9 (15.0)	0.3	23 (23.0)	1 (2.0)	0.002
1	88 (58.7)	57 (63.3)	31 (51.7)		59 (59.0)	29 (58.0)	
2	34 (22.7)	16 (17.8)	18 (30.0)		17 (17.0)	17 (34.0)	
3	4 (2.7)	2 (2.2)	2 (3.3)		1 (1.0)	3 (6.0)	
Diabetes							
No	129 (86.0)	82 (91.1)	47 (78.3)	0.03	89 (89.0)	40 (80.0)	0.143
Type 2	21 (14.0)	8 (8.9)	13 (21.7)		11 (11.0)	10 (20.0)	
History of smoking							
Smokers	21 (14.0)	19 (21.1)	2 (3.3)	0.002	16 (16.0)	5 (10.0)	0.46
Non-smokers	129 (86.0)	71 (78.9)	58 (96.7)		84 (84.0)	45 (90.0)	
Coronary artery disease							
No	137 (91.3)	80 (88.9)	57 (95.0)	0.24	93 (93.0)	44 (88.0)	0.360
Yes	13 (8.7)	10 (11.1)	3 (5.0)		7 (7.0)	6 (12.0)	
Arterial hypertension							
No	37 (24.7)	26 (28.9)	11 (18.3)	0.18	33 (33.0)	4 (8.0)	0.006
Yes	113 (75.3)	64 (71.1)	49 (81.7)		67 (67.0)	46 (92.0)	
Regurgitation							
No	56 (37.3)	35 (38.9)	21 (35.0)	0.73	45 (45.0)	11 (22.0)	0.007
Yes	94 (62.7)	55 (61.1)	39 (65.0)		55 (55.0)	39 (78.0)	
Atherosclerosis thrombosis							
No	141 (94.0)	85 (94.4)	56 (93.3)	1	94 (94.0)	47 (94.0)	1.000
Yes	9(6.0)	5 (5.6)	4 (6.7)		6 (6.0)	3 (6.0)	
Stroke							
No	142 (94.7)	85 (94.4)	57 (95.0)	1	97 (97.0)	45 (90.0)	0.118
Yes	8 (5.3)	5 (5.6)	3 (5.0)		3 (3.0)	5 (10.0)	

Note: Continuous variables are presented, mean ± SD and categorical variables as *n* (%). Abbreviations: NVAF patients, nonvalvular atrial fibrillation patients; BMI, body mass index; INR, International normalized ratio. Student’s *t*-test *p* value is marked with double asterisks (**); Mann–Whitney U test’s *p*- value is marked with single asterisk (*); The significant *p* value (*p* < 0.05).

**Table 2 genes-14-01192-t002:** Biochemical parameters of NVAF patients and comparative analysis between gender and age groups.

Characteristics	NVAF Patients, *n* = 150	Comparison between Gender			Comparison between Age Groups		
		Male, *n* = 90	Female, *n* = 60	*p* Value	Age < 65, *n* = 100	Age ≥ 65, *n* = 50	*p* Value
HGB, g/L	141.0 ± 14.5	146.6 ± 12.3	132.7 ± 13.4	0.001 **	144.8 ± 13.8	133.6 ± 12.9	0.001 **
RBC, ×10^12^/L	4.77 ± 0.66	5.00 ± 0.71	4.63 ± 0.52	0.31	4.85 ± 0.71	4.60 ± 0.51	0.002 *
PLT, ×10^9^/L	255.7 ± 59.1	253.3 ± 62.4	259.3 ± 53.9	0.18	261.7 ± 58.0	243.7 ± 59.9	0.04 *
WBC, ×10^9^/L	6.37 ± 1.76	6.52 ± 1.79	6.13 ± 1.70	0.07	6.38 ± 1.73	6.34 ± 1.84	0.91
CRE, mg/dL	0.88 ± 0.17	0.93 ± 0.16	0.79 ± 0.15	0.001 *	0.87 ± 0.16	0.88 ± 0.19	0.56
BUN, mg/dL	35.1 ± 9.70	34.8 ± 9.23	35.6 ± 10.4	0.6	34.9 ± 9.06	35.6 ± 11.0	0.69
TBIL, mg/dL	0.66 ± 0.28	0.69 ± 0.28	0.63 ± 0.28	0.13	0.66 ± 0.27	0.67 ± 0.31	0.99
DBIL, mg/dL	0.19 ± 0.16	0.17 ± 0.12	0.21 ± 0.20	0.93	0.18 ± 0.14	0.21 ± 0.20	0.5
CL, mmol/L	4.47 ± 1.06	4.38 ± 0.95	4.61 ± 1.20	0.2	4.46 ± 1.08	4.50 ± 1.01	0.84
ALT, U/L	21.4 ± 9.66	21.4 ± 8.98	21.4 ± 10.7	0.58	21.9 ± 9.42	20.2 ± 10.1	0.07
AST, U/L	21.3 ± 8.73	20.8 ± 8.08	22.1 ± 9.64	0.38	21.4 ± 8.22	21.1 ± 9.76	0.55
PT, sec	12.6 ± 3.09	12.8 ± 3.39	12.3 ± 2.60	0.53	12.6 ± 3.21	12.7 ± 2.87	0.56
PI, %	93.7 ± 22.1	92.2 ± 23.8	95.9 ± 19.3	0.33	94.2 ± 22.5	92.6 ± 21.6	0.5
APTT, sec	37.5 ± 7.19	37.2 ± 6.63	37.9 ± 8.00	0.72	37.7 ± 7.11	37.1 ± 7.42	0.53
FBG, g/L	3.20 ± 0.95	3.07 ± 0.99	3.40 ± 0.84	0.01 *	3.26 ± 1.03	3.10 ± 0.75	0.55
Dabigatran concentration, ng/mL							
Peak	104.6 ± 72.8	92.4 ± 70.3	122.8 ± 73.3	0.003 *	106.7 ± 80.7	100.5 ± 54.2	0.6
Trough	55.6 ± 37.9	49.8 ± 36.2	64.3 ± 39.2	0.002 *	53.6 ± 40.5	59.6 ± 32.3	0.05 *
Apixaban concentration, ng/mL							
Peak	164.9 ± 78.1	153.9 ± 68.9	181.4 ± 88.2	0.06	156.8 ± 73.4	181.3 ± 85.3	0.05 *
Trough	101.1 ± 45.9	92.6 ± 39.2	113.9 ± 52.2	0.01 **	93.8 ± 42.1	115.7 ± 50.1	0.002 *

Note: Continuous variables are presented, mean ± SD and categorical variables as n (%). Student’s *t*-test *p* value is marked with double asterisks (**); Mann–Whitney U test’s *p*-value is marked with single asterisk (*); The significant *p* value (*p* < 0.05); Abbreviations: NVAF patients, nonvalvular atrial fibrillation patients; HGB, hemoglobin; RBC, red blood cells; PLT, platelets; WBC, white blood cells; CRE, creatinine; BUN, blood urea nitrogen; TBIL, total bilirubin; DBIL, direct bilirubin; CL, cholesterol; ALT, alanine aminotransferase; AST, aspartate aminotransferase; PT, prothrombin time; PI, prothrombin index; APTT, activated partial thromboplastin time; FBG, fibrinogen.

**Table 3 genes-14-01192-t003:** The distributions of allelic and genotype frequencies of seven SNPs in NVAF patients and in other populations.

Gene	SNP rs Number	Genotype	NVAF Patients, *n* = 150	Allele Frequency in NVAF Patients	European	African	African-American	Asian	East Asian
*ABCB1*	rs4148738	CC	32 (21.3)	C:T = 0.43:0.57	C:T = 0.45:0.55	C:T = 0.23:0.77	C:T = 0.24:0.76	C:T = 0.40:0.60	C:T = 0.43:0.57
		CT	65 (43.3)						
		TT	53 (35.3)						
	rs1045642	AA	33 (22.0)	A:G = 0.46:0.54	A:G = 0.52:0.48	A:G = 0.22:0.78	A:G = 0.23:0.77	A:G = 0.38:0.62	A:G = 0.38:0.62
		AG	71 (47.3)						
		GG	46 (30.7)						
	rs2032582	CC	53 (35.3)	C:A = 0.53:0.47	C:A = 0.55:0.45	C:A = 0.88:0.12	C:A = 0.88:0.12	C:A = 0.43:0.57	C:A = 0.44:0.56
		CA	52 (34.7)						
		AA	45 (30.0)						
	rs1128503	AA	53 (35.3)	A:G = 0.56:0.44	A:G = 0.43:0.57	A:G = 0.21:0.79	A:G = 0.21:0.79	A:G = 0.63:0.37	A:G = 0.61:0.39
		AG	61 (40.7)						
		GG	36 (24.0)						
*CES1*	rs8192935	AA	55 (36.7)	A:G = 0.62:0.38	A:G = 0.33:0.67	A:G = 0.67:0.33	A:G = 0.67:0.33	A:G = 0.61:0.39	A:G = 0.63:0.37
		AG	76 (50.7)						
		GG	19 (12.7)						
	rs2244613	GG	37 (24.7)	G:T = 0.48:0.52	G:T = 0.19:0.81	G:T = 0.28:0.72	G:T = 0.28:0.72	G:T = 0.57:0.43	G:T = 0.61:0.39
		GT	69 (46.0)						
		TT	44 (29.3)						
	rs71647871	GG	148 (98.7)	G:A = 0.99:0.01	G:A = 0.99:0.01	G:A = 0.99:0.01	G:A = 0.99:0.01	G:A = 1.00:0.00	G:A = 1.00:0.00
		GA	2 (1.3)						
		AA	0						

Abbreviations: NVAF patients, nonvalvular atrial fibrillation patients.

**Table 4 genes-14-01192-t004:** The distributions of allelic and genotype frequencies of seven SNPs between gender groups.

Gene	SNP rs Number	Genotype	Male	Allele Frequency in Males	Female	Allele Frequency in Females	*p* Value
*ABCB1*	rs4148738	CC	21 (23.3)	C:T = 0.43: 0.57	11 (18.3)	C:T = 0.42:0.57	0.57
		CT	36 (40.0)		29 (48.3)		
		TT	33 (36.7)		20 (33.3)		
		C	78		51		
		T	102		69		
*ABCB1*	rs1045642	AA	22 (24.4)	A:G = 0.46:0.54	11 (18.3)	A:G = 0.45:0.55	0.49
		AG	39 (43.3)		32 (53.3)		
		GG	29 (32.2)		17 (28.3)		
		A	83		54		
		G	97		66		
*ABCB1*	rs2032582	CC	34 (37.8)	C:A = 0.53:0.47	19 (31.7)	C:A = 0.52:0.48	0.53
		CA	28 (31.1)		24 (40.0)		
		AA	28 (31.1)		17 (28.3)		
		C	96		62		
		A	84		58		
*ABCB1*	rs1128503	AA	33 (36.7)	A:G = 0.55:0.45	20 (33.3)	A:G = 0.57:0.43	0.46
		AG	33 (36.7)		28 (46.7)		
		GG	24 (26.7)		12 (20.0)		
		A	99		68		
		G	81		52		
*CES1*	rs8192935	AA	33 (36.7)	A:G = 0.63:0.37	22 (36.7)	A:G = 0.61:0.39	0.78
		AG	47 (52.2)		29 (48.3)		
		GG	10 (11.1)		9 (15.0)		
		A	113		73		
		G	67		47		
*CES1*	rs2244613	GG	20 (22.2)	G:T = 0.47:0.53	17 (28.3)	G:T = 0.48:0.52	0.48
		GT	45 (50.0)		24 (40.0)		
		TT	25 (27.8)		19 (31.7)		
		G	85		58		
		T	95		62		
*CES1*	rs71647871	GG	88 (97.8)	G:A = 0.99:0.01	60 (100.0)	G:A = 1.00:0.00	0.52
		GA	2 (2.2)		0		
		AA	0		0		
		G	178		120		
		A	2		0		

Note: *p* value of Hardy–Weinberg equilibrium test.

**Table 5 genes-14-01192-t005:** Comparative analysis of dabigatran and apixaban concentrations between SNP genotypes.

Gene	SNP	Genotype	Dabigatran, Peak Concentration	*p* Value	Dabigatran, trough Concentration	*p* Value	Apixaban, Peak Concentration	*p* Value	Apixaban, trough Concentration	*p* Value
*ABCB1*	rs4148738	CC	109.2 ± 74.9	0.85	60.5 ± 37.8	0.47	186.8 ± 89.3	0.22	114.4 ± 48.7	0.11
		CT	99.5 ± 68.4		53.7 ± 33.7		155.4 ± 68.4		93.7 ± 42.4	
		TT	108.1 ± 77.7		54.9 ± 43.1		163.5 ± 81.0		102.2 ± 47.3	
	rs1045642	AA	101.0 ± 73.7	0.81	54.7 ± 39.2	0.44	180.9 ± 86.3	0.51	108.1 ± 45.9	0.35
		AG	107.3 ± 77.2		56.9 ± 34.2		158.6 ± 71.9		95.5 ± 46.5	
		GG	103.1 ± 66.4		54.2 ± 43.2		163.3 ± 81.3		104.7 ± 44.9	
	rs2032582	CC	107.3 ± 78.3	0.99	54.8 ± 43.2	0.52	164.0 ± 81.1	0.24	101.3 ± 46.8	0.09
		CA	102.9 ± 66.2		53.7 ± 32.4		151.8 ± 63.1		91.5 ± 41.7	
		AA	103.4 ± 74.9		58.6 ± 38.0		181.2 ± 88.3		111.9 ± 48.2	
	rs1128503	AA	104.3 ± 81.8	0.2	57.8 ± 42.2	0.19	175.1 ± 86.3	0.55	102.9 ± 48.9	0.89
		AG	114.9 ± 74.9		59.2 ± 37.5		158.7 ± 69.2		102.1 ± 47.5	
		GG	87.6 ± 50.2		46.1 ± 30.9		160.5 ± 80.5		96.7 ± 39.2	
*CES1*	rs8192935	AA	100.9 ± 59.6	0.25	52.7 ± 30.5	0.11	157.9 ± 62.1	0.49	94.6 ± 41.4	0.57
		AG	98.7 ± 72.3		51.6 ± 35.1		164.1 ± 83.7		104.9 ± 48.2	
		GG	138.8 ± 100.1		79.8 ± 57.5		188.7 ± 95.1		104.9 ± 49.4	
	rs2244613	GG	110.9 ± 61.3	0.25	55.3 ± 34.2	0.73	166.4 ± 57.3	0.83	102.1 ± 41.6	0.47
		GT	94.6 ± 68.3		50.8 ± 27.4		163.6 ± 82.6		97.3 ± 47.6	
		TT	115.0 ± 86.9		63.3 ± 52.3		165.8 ± 87.1		106.2 ± 47.2	
	rs71647871	GG	104.7 ± 73.1	0.86	55.7 ± 38.2	0.89	165.8 ± 78.2	0.14	101.4 ± 46.0	0.5
		AG	100.9 ± 62.4		50.6 ± 15.2		99.1 ± 48.4		78.9 ± 46.2	

Note: Continuous variables are presented, mean ± SD.

**Table 6 genes-14-01192-t006:** Multiple linear regression analysis for identification influence of independent factors to peak concentration level of dabigatran.

Characteristics	Unstandardized Coefficients		Standardized Coefficients	t	*p* Value	95% Confidence Interval	
	B	Std. Error	β			Lower Bound	Upper Bound
(Constant)	56.0	126.6		0.44	0.66	−194.4	306.5
*CES1*, rs8192935	23.9	11.9	0.22	2.00	0.05 *	0.26	47.7
*ABCB1*, rs1128503	−17.9	12.2	−0.19	−1.47	0.14	−42.2	6.27
*CES1*, rs2244613	−7.98	10.8	−0.08	−0.74	0.46	−29.4	13.4
*ABCB1*, rs4148738	−1.89	27.7	−0.02	−0.07	0.95	−56.7	52.9
*ABCB1*, rs1045642	4.69	15.6	0.05	0.30	0.76	−26.2	35.6
*ABCB1*, rs2032582	−5.17	18.9	−0.06	−0.27	0.79	−42.7	32.4
*CES1*, rs71647871	15.4	51.7	0.02	0.29	0.77	−86.9	117.7
Age, years	−0.33	0.64	−0.05	−0.51	0.61	−1.58	0.93
BMI, (kg/m)	−3.23	1.71	−0.16	−1.89	0.06	−6.61	0.14
Gender (male, female)	28.8	13.8	0.19	2.09	0.04 *	1.52	56.1
APTT, sec	2.55	0.97	0.25	2.64	0.009 *	0.64	4.47
PT, sec	−2.76	2.23	−0.12	−1.24	0.22	−7.17	1.65
FBG, g/L	−0.14	6.54	−0.002	−0.02	0.98	−13.1	12.8
ALT, U/L	0.77	0.81	0.10	0.95	0.34	−0.83	2.36
AST, U/L	0.06	0.87	0.007	0.07	0.95	−1.67	1.78
CRE, mg/dL	20.4	40.2	0.05	0.51	0.61	−59.2	99.9

Note: Continuous variables are presented, mean ± SD and categorical variables as n (%). The significant *p* value (*p* < 0.05); Abbreviations: BMI, body mass index; APTT, activated partial thromboplastin time; PT, prothrombin time; FBG, fibrinogen; ALT, alanine aminotransferase; AST, aspartate aminotransferase; CRE, creatinine.

## Data Availability

The data underlying the results presented in the study are available from the authors, phone number: +7-7172706501, mail: akilzhanova@nu.edu.kz.

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
