# Peer review of "The Distribution of the Genotypes of *ABCB1* and *CES1* Polymorphisms in Kazakhstani Patients with Atrial Fibrillation Treated with DOAC"

_genes, 2023, doi:10.3390/genes14061192_

Round 1
Reviewer 1 Report
This manuscript brings up very interesting topic and points out to the usage of DOACs in a practical way, adjusting the doses prescribed according to the genetical background of NVAF patients (so-called precision medicine’s approach), in order to make the treatment more effective. Minor corrections of English language are required. There are some comments below.
1) In the Abstract paragraph, lane 21, “DOAC treatments” should be corrected to “DOACs” since treatments could not be found in plasma but drugs or their metabolites.
2) In lanes 26 through 399, Tables 2 and 5, “trough plasma concentration” probably should be corrected to “true plasma concentration”. ?
3) In the Introduction section, lane 39, “in the worldwide” probably should be corrected to “worldwide”, and “AF is affected” corrected to “AF affects”.
4) ABCB1 and CES1 gene names should be written in italics (lanes 34, 73-384 and further, and in Tables as well).
5) In Table 1 title, lane 173, what is “NUAF”? did you mean “NVAF”? I would suggest to unify the use of terms: “NVAF patients” instead of “AF patients”.
I mentioned not all grammar mistakes, there are more of them.
Minor corrections of English language are required.
I mentioned not all grammar mistakes, there are more of them.
Author Response
Response to Reviewer 1 Comments
This manuscript brings up very interesting topic and points out to the usage of DOACs in a practical way, adjusting the doses prescribed according to the genetical background of NVAF patients (so-called precision medicine’s approach), in order to make the treatment more effective. Minor corrections of English language are required. There are some comments below.
We are greatful for your review of our manuscript and valuable comments. Thank you very much! We carefully looked through paper and made corrections.
Point 1: 1) In the Abstract paragraph, lane 21, “DOAC treatments” should be corrected to “DOACs” since treatments could not be found in plasma but drugs or their metabolites.
Response 1: We understood our mistake. Thank you. We made appropriate correction.
Point 2: 2) In lanes 26 through 399, Tables 2 and 5, “trough plasma concentration” probably should be corrected to “true plasma concentration”. ?
Response 2: Thank you for the comment. Using “trough plasma concentration” we meant is the lowest concentration of a drug in the plasma. The trough level is the lowest concentration in the patient's bloodstream.
Trough Concentration, Trough Level. A pharmacokinetic measure used to determine drug dosing. Cmin is the lowest concentration of a drug in the blood, cerebrospinal fluid, or target organ after a dose is given. Peak and trough levels are drawn to determine a drug's concentration within the system. They help determine if a drug is in a toxic range or if the dosage of the medication needs to be increased. It is important to know which medications need to be monitored and what the signs and symptoms of toxicity are.
Point 3: 3) In the Introduction section, lane 39, “in the worldwide” probably should be corrected to “worldwide”, and “AF is affected” corrected to “AF affects”.
Response 3: We understood our mistake. Thank you. We made appropriate corrections.
Point 4: 4) ABCB1 and CES1 gene names should be written in italics (lanes 34, 73-384 and further, and in Tables as well).
Response 4: Thank you for comment. We made appropriate corrections.
Point 5: 5) In Table 1 title, lane 173, what is “NUAF”? did you mean “NVAF”? I would suggest to unify the use of terms: “NVAF patients” instead of “AF patients”.
I mentioned not all grammar mistakes, there are more of them.
Response 5: Thank you for comment. We made appropriate corrections.

Reviewer 2 Report
In the present study, Abdrakhmanov et al. aim at investigating y the distribution and influence of polymorphisms of the genes encoding P-glycoprotein (ABCB1) and carboxylesterase 1 (CES1) on variability of plasma concentrations of DOAC treatment in Kazakhstani patients with NVAF. They conclude that polymorphism in CES1 gene was significantly associated with plasma concentrations of dabigatran in Kazakhstani AF patients.
The study might be of interest for the readers but the evaluation of gene polymorphisms in real world appear to be limited. Hence, authors should indicate who are the patients in which this evaluation should be suggested. Practically, in case of ischemic event in patient already treated with a DOAC, it is common practice to change with another DOAC
Rivaroxaxan and Edoxaban are not considered for analysis, please explain why
Please, check all the article for typos (from line 81 to line 380). There are error in grammar and mean of sentences
Author Response
Response to Reviewer 2 Comments
In the present study, Abdrakhmanov et al. aim at investigating the distribution and influence of polymorphisms of the genes encoding P-glycoprotein (ABCB1) and carboxylesterase 1 (CES1) on variability of plasma concentrations of DOAC treatment in Kazakhstani patients with NVAF. They conclude that polymorphism in CES1 gene was significantly associated with plasma concentrations of dabigatran in Kazakhstani AF patients.
The study might be of interest for the readers but the evaluation of gene polymorphisms in real world appear to be limited. Hence, authors should indicate who are the patients in which this evaluation should be suggested. Practically, in case of ischemic event in patient already treated with a DOAC, it is common practice to change with another DOAC.
We are greatful for your review of our manuscript and valuable comments. Thank you very much! We carefully looked through paper and made corrections.
Pharmacogenomics could be useful to predict the better clinical response and avoid adverse events in patients treated with anticoagulants, identifying the most appropriate anticoagulant drug for each patient. Available DOACs for NVAF and venous thromboembolism include dabigatran, a selective anti‐factor IIa molecule, and three direct anti‐factor Xa inhibitors: apixaban, edoxaban, and rivaroxaban. They have stable pharmacokinetics and do not require routine blood checks, thus avoiding most of the drawbacks of VKAs.
Current pharmacogenomics data show that the polymorphisms affecting VKAs or DOACs are different, concluding that personalized medicine based on pharmacogenomics could be possible.
We agree that the DOACs’ pharmacological characteristics, together with the assumed predictable dose‐response, led to the indication of fixed dose administration without dose adjustment based on laboratory testing [1]. The choice of DOAC dosage is based on the evaluation of clinical indications (NVAF, venous thromboembolism), patient characteristics (age, gender, body weight, concomitant administration of potentially interfering drugs), and renal and liver function, assuming that drug anticoagulant effect is prevalently controlled by these conditions.
Nevertheless, a high interindividual variability in the drug blood levels was shown with all DOACs, and post hoc analyses of phase III trials showed an association between DOAC plasma levels and thrombotic and bleeding complications during follow up [2, 3, 4, 5, 6, 7, 8, 9]. Moreover, phase IV clinical trials have shown a higher interindividual variability if compared with phase III studies, confirming that real world patients differ from the selected populations enrolled in randomized trials.5, 6, 7, 8, 9
More studies are required to implement personalized medicine in clinical practice with OA and based on pharmacogenetics of DOACs specially in different ethnical groups.
Currently, DOACs represent the first‐line treatment in two clinical conditions: the prevention of stroke and systemic embolism in patients with NVAF and the treatment/prevention of venous thromboembolism.1
- Ageno W, Gallus AS, Wittkowsky A, Crowther M, Hylek EM, Palareti G.Oral Anticoagulant Therapy: Antithrombotic Therapy and Prevention of Thrombosis, 9th ed.: American College of Chest Physicians Evidence‐Based Clinical Practice Guidelines. Chest. 2012;141:e44S–88S. [PMC free article] [PubMed] [Google Scholar]
- (FDA) FaDA .Pradaxa ‐ clinical pharmacology and biopharmaceutics review(s). 2010. https://www.accessdata.fda.gov/drugsatfda_docs/nda/2010/022512Orig1s000ClinPharmR_Corrrected%203.11.2011.pdf(accessed 17 January 2019). https://wwwaccessdatafdagov/drugsatfda_docs/nda/2010/022512Orig1s000ClinPharmR_Corrrected%203112011pdf. 2010.
- (FDA) FaDA .Xarelto ‐ clinical biopharmaceutics review(s). 2011. http://www.accessdata.fda.gov/drugsatfda_docs/nda/2011/022406Orig1s000ClinPharmR.pdf (accessed 26 September 2017). 2011.
- (FDA) FaDA .Eliquis ‐ clinical pharmacology and biopharmaceutics review(s). 2012. http://www.accessdata.fda.gov/drugsatfda_docs/nda/2012/202155Orig1s000ClinPharmR.pdf Accessed 26 September 2017.
- (EMA) EMA .Pradaxa ‐ summary of product characteristics. http://www.ema.europa.eu/docs/en_GB/document_library/EPAR_-_Product_Information/human/000829/WC500041059.pdf. Accessed 26 September 2017.
- (EMA) EMA .Xarelto ‐ Summary of product characteristics. http://www.ema.europa.eu/docs/en_GB/document_library/EPAR_-_Product_Information/human/000944/WC500057108.pdf. Accessed 26 September 2017.
- (EMA) EMA .Eliquis ‐ Summary of product characteristics. http://www.ema.europa.eu/docs/en_GB/document_library/EPAR_-_Product_Information/human/002148/WC500107728.pdf. Accessed 26 September 2017.
- (EMA) EMA .Lixiana ‐ summary of product characteristics. European Medicines Agency. Lixiana ‐ summary of product characteristics. 2017. http://www.ema.europa.eu/docs/en_GB/document_library/EPAR_-_Product_Information/human/002629/WC500189045.pdf. Accessed 26 September 2017.
- Testa S, Tripodi A, Legnani C, Pengo V, Abbate R, Dellanoce C, et al.Plasma levels of direct oral anticoagulants in real life patients with atrial fibrillation: results observed in four anticoagulation clinics. Thromb Res. 2016;137:178–83. [PubMed] [Google Scholar]
Point 1: Rivaroxaxan and Edoxaban are not considered for analysis, please explain why
Response 1: Thank you for comment.
In Kazakhstan, only generic drugs are allocated as rivaroxaban and edoxaban drugs for free inpatient care. Evidence on efficacy and safety in patients with atrial fibrillation, supported by both randomized clinical trials and studies in real clinical practice, exists only for the original rivaroxaban. It is noted that generics produced by different companies differ significantly in bioavailability, and as a result, they can lead to a change in their therapeutic effect and distortion of the study results. Therefore, rivaroxaban and edoxaban were not included in the study.
Point 2: Please, check all the article for typos (from line 81 to line 380). There are error in grammar and mean of sentences
Response 2: Thank you for comment. We made appropriate corrections through all the text.

Round 2
Reviewer 2 Report
Authors have addressed the raised issues